# Brief Screening for Distress among Healthcare Professionals: Psychometric Properties of the Physician Well-Being Index—Spanish Version

**DOI:** 10.3390/ijerph19159451

**Published:** 2022-08-02

**Authors:** Rebeca Robles, Ana Fresán, Natasha Alcocer-Castillejos, Janet Real-Ramírez, Silvia Morales-Chainé

**Affiliations:** 1Global Mental Health Research Center, Ramón de la Fuente Muñiz National Institute of Psychiatry, Mexico City 14370, Mexico; 2Laboratory of Clinical Epidemiology, Subdirectorate of Clinical Research, Ramón de la Fuente Muñiz National Institute of Psychiatry, Mexico City 14370, Mexico; a_fresan@yahoo.com.mx; 3Neurology and Psychiatry Department, Salvador Zubirán National Institute of Medical Sciences and Nutrition, Mexico City 14080, Mexico; natdian@gmail.com; 4School of Public Health of Mexico, National Institute of Public Health, Mexico City 62100, Mexico; janet.real@insp.mx; 5Psychology Faculty, National Autonomous University of Mexico, Mexico City 04510, Mexico; smchaine@gmail.com

**Keywords:** healthcare workers, mental health, distress, evaluation, Well-being Index

## Abstract

Background: The Physician Well-Being Index (PWBI) is a brief, valid, reliable self-assessment instrument to identify health professionals’ distress and those in need of an intervention. Objective: to evaluate the construct, predictive validity (of depression, suicidal ideation, insomnia, and generalized anxiety), and internal consistency of the 7-item Spanish version of the PWBI (PWBI-S). Methods: out of a national population of approximately 1 million Mexican healthcare professionals, a sample of 3506 subjects (42.0% physicians, 28.7% nurses and 29.3% psychologists) completed an online survey between 17 April and 7 May 2020, at the time of the COVID-19 case cluster transmission scenario in Mexico. Results: In the three sub-samples, PWBI-S’s Confirmatory factor analyses (adding residual covariances) exhibited adequate goodness of fit indices for the PWBS original unidimensional model. Overall Cronbach’s alphas were 0.89 for physicians, 0.90 for nurses, and 0.86 for psychologists. Univariate logistic regression models showed that a cutoff point of 3 on the total score of the PWBI-S was generally related to the presence of depression, suicidal ideation, and insomnia, but not with generalized anxiety among nurses and psychologists. When trying with a cutoff point of 3, a relationship with GA was shown in psychologists, but not in nurses. Conclusions: our findings suggest that PWBI-S is a valid, reliable measure for clinical and research purposes in the field.

## 1. Introduction

The advent of the COVID-19 pandemic and its enormous psychological impact on frontline healthcare workers [1] has dramatically increased interest in the mental well-being of health professionals, both, because of the high levels of burnout previously reported in this group and for its implications over the quality of care they provide for patients [2]. Health professionals’ psychological distress, defined as a state of emotional suffering associated with stressors and demands that are difficult to address in daily life [3], has been associated with malpractice [4], attrition from clinical practice [5], and mental health problems, including suicidal ideation [6]. Health professionals’ lack of awareness of or reluctance to acknowledge personal distress constitutes a significant barrier to seeking help when required [7].

A valid and reliable tool is needed to increase the systematic screening and identification of and early intervention in health professionals’ psychological distress. However, psychological distress has generally been assessed using screening measures for depression, anxiety, and stress. Although there are valid, reliable instruments to evaluate these mental health problems, psychological distress can be experienced independently of these factors [8]. Other instruments often employed to evaluate distress in the workforce are those developed to evaluate burnout, which are usually long tools measuring only one or two dimensions of distress; some address job satisfaction/meaning, others engagement at work, fatigue, or sleepiness [7,9].

To address these problems, Dyrbe et al. [7] developed the Physician Well-Being Index (PWBI), a brief, valid, reliable self-assessment instrument designed to identify healthcare workers in severe distress probably requiring intervention. The PWBI has been validated in medical students, medical residents, and general workers in the USA. Ratings above the proposed cutoff have been related to risk of poor mental quality of life, burnout, severe fatigue and suicidal ideation (all considered a dimension of well-being), and are also useful for identifying physicians in risk of negative impacts such as low career satisfaction, a medical error and those contemplating leaving their current position [10]. A detailed description of its content, rating scale, interpretation, and psychometric properties is provided in the measures subsection below.

The present study was designed to test the psychometric properties of the 7-item Spanish version of the PWBI (PWBI-S) in Mexican healthcare professionals. The assessment was undertaken during the COVID-19 case cluster transmission scenario in Mexico to verify the validity and reliability of PWBI-S for its use as a screening measure of distress along a health emergency in three different populations: physicians, nurses, and psychologists. Confirmatory factor analyses (CFA) and invariance measurement were performed, and internal consistency and predictive validity was determined. 

## 2. Materials and Methods

This was a cross-sectional survey of physicians, nurses and psychologists >18 years old currently residing in Mexico, performed between 17 April and 7 May 2020. According to Comrey and Lee suggestion of 1000 or more subjects as an excellent sample size for CFA [11], we stop recruiting participants for this study when at least 1000 individuals for each subsample (physicians, nurses, and psychologists).

### 2.1. Measures

*The Physician Well-Being Index—Spanish version (PWBI-S):* This instrument was developed as a brief (7-item) tool capable of identifying individuals whose distress places them at risk for serious, adverse, life-threatening mental health consequences. The seven items are representative of manifestations of burnout, depression, fatigue, stress, and poor quality of life. All questions are answered using a simple yes/no format. One point is assigned for each “yes” answer with summary scores on the 7-item index ranging from 0 to 7. Thus, higher scores reflect higher levels of experienced distress [12]. The instrument has proved its utility in medical students as well as physicians [7]. Several cutoff points have been used (particularly 3, 4, and 5) to identify a subset of individuals whose degree of distress could be more likely to contribute to additional negative consequences. MedEd Web Solutions (MEWS) and the Mayo Clinic provided the original English version and the guidelines for its use for research purposes (Well-being Index Research document) and authorized this first translation into Spanish. According to the Sperber suggestions for the translation of instruments for cross-cultural research [13], this process was made by a Mexican bilingual mental health professional in order to avoid a literal translation and devote enough attention to cultural nuances, particularly to: (1) idiomatic expression (e.g., burnout, which was not translated literally or technically, but with terms generally used by Mexican health professionals to express their work-related psychological and emotional fatigue: “*desgastado o agotado por el trabajo*”—which would be literally translated as: “worn or exhausted by work”), and (2) emotionally evocative terms (e.g., hardening you emotionally, which was translated using the terms that are more understandable or better to describe this dimension of burnout and emotional distress in Mexican colloquial language: “*lo ha vuelto frío o distante”*—which would be literally translated as: “has made you cold or distant”).

For the present study, it was decided to prove the cutoff points 3, 4, and 5 of the PWBI-S to determine which was more adequate to determine predictive validity for increased risk of depression, suicidal ideation, insomnia, and generalized anxiety. 

*The Patient Health Questionnaire-2 (PHQ-2), Spanish version*: An ultra-brief, two-item screening tool for depression that assesses frequency of depressed mood and anhedonia over the past 2 weeks using a 4-point scale ranging from 0 = not at all to 3 = nearly every day. Scores range from 0 to 6. A cutoff point of 3 was used to dichotomize the score in those at no or minimal risk for depression (<3) and individuals at moderate to high (≥3) risk for depression [14]. The Spanish version of the PHQ-2 used in the present study has demonstrated adequate psychometric properties in the Mexican population [15].

For the evaluation of suicidal ideation, the yes/no item *“In the past month, have you felt that you wanted to die, or thought about being dead?”* from the Spanish version of the five-item *Depression scale from the*
*ICD-11 PHC field study* was used [16], Thus, in the present study, participants were considered to have suicidal ideation when answering affirmatively to this question.

For insomnia, we employed the question that assesses *“how much have been bothered by troubles falling or staying asleep?”* from the *Spanish version of the PTSD Checklist for DSM-5* [17], which has shown adequate psychometric properties in Mexican mental health service users [18]. Respondents rated the frequency of the symptom over the last month on a 5-point scale ranging from 0 = not at all to 4 = extremely. Thus, in the present study, participants were considered to have insomnia when they selected 3 or 4 as their answer to this question. 

The *Spanish version of the five-items Anxiety Scale from the ICD-11 PHC* was used to assess generalized anxiety. In this scale, all items are answered using a simple yes/no format. One point is assigned for each “yes” answer; thus, the standard scoring range from 0 to 5; a cutoff point of 3 is considered adequate for the correct identification of Mexican individuals with generalized anxiety [6].

### 2.2. Procedures

Healthcare professionals were invited through official media (i.e., federal government’s microsite coronavirus.gob.mx, press conferences, and the National Institute of Psychiatry’s website and social networks) to participate in an anonymous online survey developed by the “Ramón de la Fuente Muñíz” National Institute of Psychiatry, Ministry of Health, in collaboration with the National Autonomous University of Mexico (Ciudad de Mexico, Mexico). 

At the beginning of the survey, demographic and work-related features were assessed. Then, all participants completed the instruments. At the end of the survey, participants received brief personal feedback regarding their punctuations in the scales and when they indicate high risk of burnout and/or the other mental health problems evaluated, adding specific contact information for those requiring specialized mental health treatment.

### 2.3. Data Analyses

Descriptive statistics were used to characterize the sample. Chi-squared tests and one-way ANOVA with Bonferroni correction were used to compare demographic, professional and health variables between physicians, nurses, and psychologists. To determine the construct validity of the PWBI-S we performed several steps including confirmatory factor analysis (CFA), multi-group confirmatory factor analysis (multi-group CFA) and the measurement of invariance to test the appropriateness of the one-dimensional model originally proposed for the PWBI [9] in these three subsamples. For the CFA and multi-group CFA, the average lower standardized loading factors (standardized regression weighted estimates) included in the model should be at least 0.40, as this value and over suggests that the indicator variables (items) are adequate and representative of their latent variable (PWBI-S in this particular study) [19]. Based on Hatcher recommendations [20], maximum likelihood estimation with standardized coefficients and values was the method used. The model’s goodness of fit was borne out by a chi-square ratio (χ^2^/df) with acceptable values near or lowers than 3.0 [21]. However, other recommended criteria for goodness of fit were included, such as a Root Mean Square Error of Approximation (RMSEA) with adequate values lower than 0.05 for good fit, between 0.05 and 0.08 as acceptable, and between 0.08 to 0.10 as marginal [22], Bentler’s Comparative Fit Index (CFI) and the Tucker–Lewis index (TLI), both with values greater than 0.95 and the Standardized Root Mean-Square Residual (SRMR) with adequate values lower than 0.08 [23]. First, a CFA using the whole sample was tested. Then, modification indices with the most important reductions in chi-square value were added to the model. With this initial model, a multi-group CFA was performed. Second, the model obtained in the multi-group CFA was tested separately for each group (physicians, nurses, and psychologists) by using CFA with maximum likelihood estimation. According to Vandenve and Lance suggestions [24], several steps with the multi-group CFA were used to test invariance: to examine configural invariance, the multi-group CFA was examined separately (physicians, nurses, and psychologists) using CFA. If the same model fits adequately between groups, configural invariance is supported. Then we examined metric invariance by comparing the ΔCFI (the difference in CFI between the initial multi-group CFA and the models for each sample). A ΔCFI less than 0.01 indicate invariance [25]. We did not use the chi-square difference test as it has been shown that it is highly sensitive to sample size and less sensitive than the ΔCFI. After CFAs and invariance procedures were performed, the internal consistency of the models in each group (using the PWBI-S items) was tested with Cronbach’s alpha, considering an item-total correlation greater than 0.3 as an indicator of the relation of the item with the overall scale and an alpha value equal to or greater than 0.80 as an adequate reliability indicator [26]. These analyses were performed with the Stata/SE 13.0 software for Windows. For the predictive validity of the PWBI-S, we performed univariate logistic regression models using the three proposed cutoff points (3, 4, and 5) from the PWBI (Well-being Index Research document guideline). Each cutoff point was included as an independent variable to determine which score is better for identifying higher risks for presenting adverse mental health outcomes (depression, suicidal ideation, insomnia, and generalized anxiety) in our sample. We identify the best cutoff point in accordance to: (a) having a significant odd ratio (OR) and (b) the narrowest (95%) confidence interval of the significant OR. If more than one cutoff point exhibited adequate statistical indices in the logistic regression analysis, the lowest cutoff point was selected as the one related to the development of adverse mental health outcomes. After these analyses were performed, we then compared the PWBI-S total score between groups.

## 3. Results

### 3.1. Sample Description, Health Conditions and PWBI-S

Out of a national population of 1,085,266 health care professionals (270,600 general practitioners [27], 22,613 medical residents [28], 147,910 medical specialists [29], 315,000 nurses [30], and 329,143 psychologists [31]), a total of 3506 healthcare professionals participated in the survey, of which 42.0% (*n* = 1473) were physicians, 28.7% (*n* = 1006) nurses, and the remaining 29.3% (*n* = 1027) psychologists. Most participants were women (75.6%, *n* = 2651), mostly partnered (52.7%, *n* = 1849), with a mean age of 42.7 years (S.D. = 11.0), with physicians being older than nurses (Bonferroni < 0.001) and psychologists (Bonferroni < 0.001) (see Table 1). At the time of the survey, more than half the healthcare professionals were working at a COVID-19 center (61.3%, *n* = 2150) with a higher frequency of nurses and physicians working at these institutions. A quarter (*n* = 881) were frontline healthcare professionals, mostly physicians and nurses (Table 1).

As can be seen in Table 1, generalized anxiety was the most frequent health condition reported by all healthcare professionals (at least 70% in each group), followed by insomnia, reported by 29.0% (*n* = 1015), depression (27.8%, *n* = 975) and suicidal ideation (12.5%, *n* = 439). Insomnia, depression, and suicidal ideation were more frequently reported in physicians while generalized anxiety was less reported by nurses. 

### 3.2. Construct Validity, Invariance Measurement and Internal Consistency of the PWBI-S

The first CFA performed with the whole sample (model 1) displayed inadequate goodness-of-fit indices; modification indices (MI) suggested residual covariances (item 1 with item 2 and 3; item 2 with item 3 and 4; item 3 with item 6; item 4 with item 7; and item 5 with item 7). Although some of these residual covariances were <0.1, they all improved the model with adequate goodness-of-fit indices (model 2), suggesting that the unidimensional model with seven items originally proposed is a stable, valid construct for all health professionals of the study (see Table 2 and Figure 1). With this second model, a multi-group CFA was tested (model 3) showing acceptable RMSEA values and adequate values in the remaining indices. After showing the stability of the model with the multi-group CFA, it was tested for physicians, nurses, and psychologists separately (models 4.1, 4.2, and 4.3). As in the multi-group CFA, the independent models showed acceptable RMSEA values with the remaining indices being adequate (Table 2 and Figure 2). For all the models, item loadings were higher than 0.40, with the lowest item load found in model 2 with a 0.57 value. Nor in the multi-group analysis or the independent CFA models for physicians, nurses, and psychologists were modification indices added.

In terms of invariance measurement, the results of the CFA models (3 and 4) showed that the same measurement model fits adequately to the data across physicians, nurses, and psychologists, which supports configural invariance. When metric invariance was tested by comparing the ΔCFI, all values were less than 0.01 (multigroup CFA vs. physicians = 0.003; multigroup CFA vs. nurses = 0.007; and multigroup CFA vs. psychologists = 0.005), a result that supports invariance.

Reliability analysis showed that the overall Cronbach’s alpha values of the PWBI-S were adequate: 0.90 for all mental health professionals (item-total correlations between 0.56 and 080); 0.89 for physicians (item-total correlations between 0.54 and 0.80), 0.90 for nurses (item-total correlations between 0.58 and 0.80), and 0.86 for psychologists (item-total correlations between 0.58 and 0.80).

### 3.3. Predictive Values of the PWBI-S and Group Comparison of the PWBI-S Total Score

Finally, the results of the univariate logistic regression models showed that the cutoff point of 3 on the total score of the PWBI-S was the most adequate for identifying increased risks for the development of depression, suicidal ideation, and insomnia in physicians, nurses, and psychologists. The regression models were not completely clear for generalized anxiety. As shown in Table 3, the cutoff point of 4 for physicians and a cutoff point of 3 for psychologists could be the most adequate for identifying an increased risk for generalized anxiety. Nevertheless, the OR confidence intervals were too broad for the cutoff points of 3 and 4 for nurses, while the cutoff of 5 was not significant for nurses or psychologists.

The mean total score of the PWBI-S differed among groups (F = 83.4, *p* < 0.001, Bonferroni < 0.001 for post hoc tests). Physicians reported the highest score, followed by nurses and psychologists.

## 4. Discussion

The principal aim of the present study was to determine the construct and predictive validity and the internal consistency of the Spanish version of the Physician Well-Being Index (PWBI-S) to have a brief, adequate self-assessment instrument to evaluate heath professionals’ distress and identify those in need of an intervention. In this regard, our findings suggest that PWBI-S is a valid, reliable measure for evaluating the level of psychological distress of three major health professional groups: physicians, nurses, and psychologists.

This has at least three significant implications. First, the original version of the PWBI in English has proven to have adequate psychometric properties among medical students, physicians, and residents [7,9,10,32], as well as nurse practitioners and physician assistants [33]. The present study provides further evidence of its utility in another key group of health professionals: psychologists.

Second, although translations of the PWBI into other languages and psychometric evaluations of subjects in various contexts have been published—such as Malaysian medical students [34]—this is the first study to evaluate the version in Spanish, one of the most widely spoken languages in the world. In congruence with the previous studies using a strong methodology to evaluate the construct validity of PWB´s translation into another language [35], we use CFA and MI, residual covariances and standardized regression weighted as indicators to select which items were fit to be kept in the final model for each group of health professionals [26], as well as Cronbach’s alpha coefficients to determine internal consistency values, which were comparable with previous studies reporting coefficients higher than 0.6 [12,34], Furthermore, we used a very large sample, which allows the development and evaluation of a model for each subsample (physicians, nurses, and psychologists).

Third, our data provide evidence of the validity of the PWBI’s cutoff scores for the most common mental health problems among different groups of healthcare professionals experiencing high levels of work-related stress in a middle-income country (in this case due to the COVID-19 pandemic) [35]. Generally, healthcare professionals with more than 3 points should be further evaluated and, if necessary, appropriate support should be provided to improve their psychological well-being.

According to Shanafelt et al. [36], use of the PWBI could enhance the ability of health professionals to become aware of their level of distress and contemplate behavioral changes to improve their well-being, which is also an important prerequisite for seeking and maintaining mental health treatment, if required. In this respect, the present study could have a heuristic to improve the evaluation of distress among Spanish-speaking healthcare professionals and therefore the treatment and alleviation of mental health consequences and ultimately the improvement of their well-being.

Although the sample size and selection are appropriate for a psychometric study using CFA (requiring large numbers of subjects), it is not representative of all Mexican healthcare workers. Those who did not have Internet access were obviously excluded, and there is a potential selection bias of respondents who may experience more symptoms than non-respondents (given that the evaluation was promoted as a tool for identifying mental disorders and referring subjects to mental health treatment if needed).

Moreover, indicators of predictive validity were assessed using self-reported screening measures rather than psychiatric interviews (considered the gold standard for the diagnosis of mental disorders). The results of this study should therefore not be taken as estimates of prevalence or other epidemiological parameters of distress and mental health problems among health professionals coping with COVID-19 in Mexico.

Regarding the psychometric evaluation of PWBI, although it has proven to be a stable measure at different times [33], future research should examine its sensitivity to change when interventions to prevent and reduce psychological distress among health professionals are implemented. Furthermore, continued research is required to refine and verify its psychometric properties in other key healthcare workers.

## 5. Conclusions

The present study demonstrated that PWBI-S is a brief, valid, reliable screening instrument to assess psychological distress in Mexican physicians, nurses, and psychologists coping with a stressful health emergency (such as COVID-19), which could therefore be used for clinical and research purposes in the field. A score of 3 points in the PWBI-S could be regarded as indicative of significant psychological distress.

## Figures and Tables

**Figure 1 ijerph-19-09451-f001:**
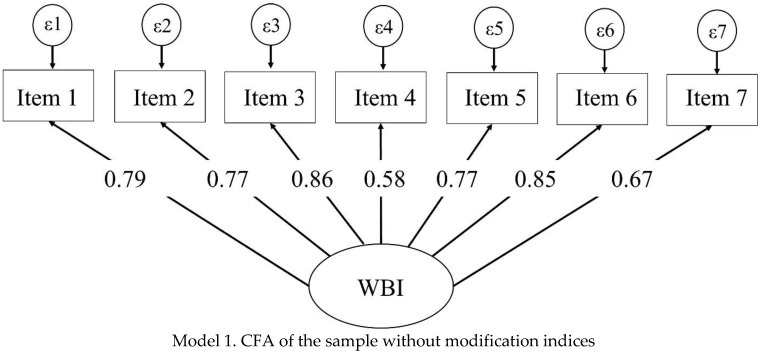
Confirmatory factor analysis of the total sample without modification indices (**Model 1**) and after modification indices were added (**Model 2**).

**Figure 2 ijerph-19-09451-f002:**
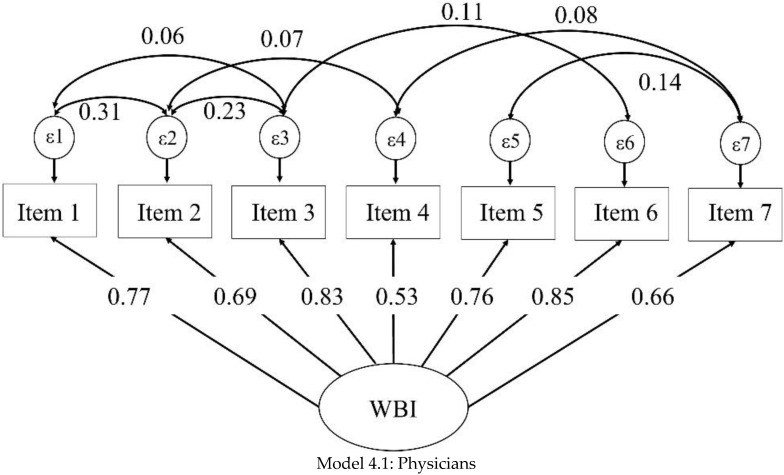
Multigroup confirmatory factor analysis: physicians (**Model 4.1**), nurses (**Model 4.2**), and psychologists (**Model 4.3**).

**Table 1 ijerph-19-09451-t001:** Sample description.

	Total *n* = 3506	Physicians *n* = 1473	Nurses *n* = 1006	Psychologists *n* = 1027	Statistic
Demographic features: *n* (%) or Mean + S.D					
Sex—Women	2651 (75.6)	928 (63.0)	890 (88.5)	833 (81.1)	χ^2^ = 234.0, *p* < 0.001
Age—Years	42.7 ± 11.0	44.2 ± 11.3	41.1 ± 10.6	42.2 ± 10.9	F = 26.0, *p* < 0.001
Marital status—Partnered	1849 (52.7)	801 (54.4)	525 (52.2)	523 (50.9)	χ^2^ = 3.06, *p* = 0.21
Professional features: *n* (%)					
Institution type—COVID-19 center	2150 (61.3)	986 (66.9)	689 (68.5)	475 (46.3)	χ^2^ = 139.7, *p* < 0.001
Frontline—Yes	881 (25.1)	503 (34.1)	305 (30.3)	73 (7.1)	χ^2^ = 255.3, *p* < 0.001
Mental health conditions: *n* (%)					
Generalized anxiety—Yes	2649 (75.6)	1163 (79.0)	705 (70.1)	781 (76.0)	χ^2^ = 25.8, *p* < 0.001
Depression—Yes	975 (27.8)	499 (33.9)	265 (26.3)	211 (20.5)	χ^2^ = 57.0, *p* < 0.001
Suicidal ideation—Yes	439 (12.5)	228 (15.5)	98 (9.7)	113 (11.0)	χ^2^ = 21.0, *p* < 0.001
Insomnia—Yes	1015 (29.0)	499 (33.9)	271 (26.9)	245 (23.9)	χ^2^ = 32.3, *p* < 0.001

Generalized Anxiety—cutoff point ≥ 3 on the Anxiety Scale from the ICD-11 PHC; Depression—cutoff point ≥ 3 on the PHQ-2; Suicidal ideation—yes/answer; Insomnia—cutoff point ≥ 3 on the PCL-14 item.

**Table 2 ijerph-19-09451-t002:** Results of confirmatory factor analyses.

Sample	Chi-Square Ratio (χ^2^/df)	Goodness-of-Fit Indices				
		RMSEA	90% C.I.	CFI	TLI	SRMR
Model 1: All health professionals	31.95	0.094	0.087–0.102	0.969	0.954	0.02
Model 2: Model 1 + MI	5.63	0.036	0.026–0.048	0.998	0.993	0.008
Model 3: Multi-group CFA	5.16	0.060	0.052–0.067	0.987	0.981	0.02
Model 4: CFA for each group						
4.1 Physicians	4.81	0.051	0.034–0.069	0.995	0.986	0.01
4.2 Nurses	6.19	0.072	0.052–0.093	0.991	0.974	0.01
4.3 Psychologists	5.12	0.063	0.044–0.085	0.993	0.979	0.01

RMSEA—Root Mean Square Error of Approximation; 90% C.I—Confidence interval of the RMSEA; CFI—Bentler’s Comparative Fit Index; TLI—Tucker-Lewis Index; SRMR—Standardized Root Mean-Square Residual. Although the χ^2^/df ratio did not met adequacy in the models, the remaining goodness-of-fit indices were adequate (CFI, TLI and SRMR) or acceptable (RMSEA < 0.08).

**Table 3 ijerph-19-09451-t003:** Univariate logistic regression for mental health conditions using the Spanish version of the Physician Well-Being Index (PWBI-S) to determine the risk for the development of adverse mental health outcomes and comparison of the mean PWBI-S scores among groups.

Mental Health Condition (PWBI-S Cutoff Points)	Physicians			Nurses			Psychologists		
	β	OR	OR 95% C.I.	β	OR	OR 95% C.I.	β	OR	OR 95% C.I.
Generalized anxiety									
Cutoff point 3	3.42	30.50	11.30–82.62	4.36	78.35	10.90–562.78	2.35	10.52	2.55–43.33
Cutoff point 4	3.07	21.70	8.01–58.75	4.14	62.77	8.73–451.47	20.10	536,311,486	- *
Cutoff point 5	2.99	19.98	6.34–62.96	20.48	786,825,125	- *	20.08	529,170,188	- *
Depression									
Cutoff point 3	1.94	7.00	5.35–9.15	2.11	8.28	5.65–12.14	1.97	7.17	4.22–12.17
Cutoff point 4	2.17	8.82	6.48–12.00	2.16	8.73	5.76–13.24	2.73	15.44	7.22–33.0
Cutoff point 5	2.55	12.92	8.74–19.10	2.80	16.58	9.42–29.18	2.85	17.32	6.98–42.98
Suicidal ideation									
Cutoff point 3	0.76	2.13	1.57–2.90	1.74	5.73	3.66–8.98	1.09	2.97	1.62–5.44
Cutoff point 4	0.84	2.31	1.67–3.19	1.86	6.48	4.08–10.29	1.57	4.83	2.44–9.55
Cutoff point 5	0.93	2.55	1.79–3.62	2.14	8.57	5.19–14.13	1.62	5.08	2.35–10.97
Insomnia									
Cutoff point 3	1.39	4.05	3.14–5.22	2.14	8.56	5.82–12.58	1.19	3.28	1.96–5.49
Cutoff point 4	1.67	5.34	4.01–7.11	1.94	7.34	4.88–12.04	1.42	4.18	2.20–7.94
Cutoff point 5	1.88	6.57	4.69–9.21	2.25	9.57	5.79–15.79	1.48	4.41	2.11–9.21
Group comparison of the PWBI-S total score									
PWBI-S total score	Mean	S.D.	Range	Mean	S.D.	Range	Mean	S.D.	Range
	1.27	2.09	0–7	0.86	1.81	0–7	0.33	1.16	0–7

All *p*-values for the logistic regression analyses were <0.001, except for: * the analyses for General Anxiety with a cutoff point of 4 for nurses and cutoff points of 3 and 4 for psychologist, all with a *p*-value of 0.99.

## Data Availability

The data presented in this study are available on request from the corresponding author.

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
