# Peer review of "Brief Screening for Distress among Healthcare Professionals: Psychometric Properties of the Physician Well-Being Index—Spanish Version"

_ijerph, 2022, doi:10.3390/ijerph19159451_

Round 1
Reviewer 1 Report
Thanks for the opportunity to review this manuscript.
The authors developed the Spanish version of the 7-item Physician Well-Being Index (PWBI-S) and examined its psychometric qualities among physicians, nurses, and psychologists. It was found that the unidimensional model (after adding residual covariances) is applicable for the three samples. Moreover, the PWBI-S also demonstrated good internal consistency and validity for the samples respectively.
The study was well designed. The quality of the manuscript is also satisfactory. However, some writings require attention to avoid misunderstanding. Below are my comments for the authors' consideration:
A. Abstract
1. Lines 23-25, "Confirmatory factor analyses performed of the three samples exhibited adequate goodness of fit indices for the unidimensional model in the original PWBI". The sentence seems to reporting the results for the PWBI but not PWBI-S. Moreover, it is important to report that residual covariances were added.
2. Lines 26-28, "Univariate logistic regression models showed that a cutoff point of 4 on the total score of the PWBI-S was generally adequate for predicting depression, suicide ideation and insomnia, while a cutoff point of 3 was adequate for predicting generalized anxiety in psychologists."
It was clearly indicated that the cutoff point of 3 is applicable for psychologists. What about the cutoff point of 4? In addition, as a cross-sectional design was used, it is misleading to interpret the results with a lens of causality.
B. Introduction
1. The introduction of the Physician Well-Being Index (PWBI) shall be enhanced. The authors shall at least briefly review the development of the PWBI and its underlying theoretical framework (if any). It is also necessary to review past studies that examined the psychometric qualities of the PWBI. Finally, the authors shall highlight the problem statement of the study.
2. Lines 57-60, it may not be a good idea to explain how the predictive validity of the PWBI-S was tested here. Readers will experience difficulty to understand the contents before knowing the methods and variables of the study.
C. Materials and Methods
1. It is important to provide the details of the measurements such as the rating scale (e.g., 5-point Likert scale?), especially for the PWBI-S. The information will be helpful for evaluating the appropriateness of using the maximum likelihood estimation. The authors may consider introducing the measurements used in the study first and followed by the procedure of the survey.
2. Line 70, "Then, all participants completed the PWBI." It is confusing whether the participants answered the PWBI or PWBI-S.
3. Kindly indicate whether participants answered the English or Spanish version of the measurements.
4. Some abbreviations (e.g., MEWS) are not defined or introduced.
5. Development of the Spanish version of the PWBI is one of the key contributions of this study. The authors shall report more information about the translation procedure.
6. Lines 81-83, "For the present study, it was decided to use the median score of the PWBI (4 points) to determine its predictive validity for increased risk of depression, generalized anxiety, suicide ideation and insomnia."
As several cutoff points have been suggested, the authors shall justify why the use the median score, but not other scores, for testing the predictive validity.
7. Lines 92-94, " the ICD-11 PHC field study was used.12 For insomnia, the question that assesses problems falling or staying asleep from the PTSD Checklist for DSM-5 93 was used (13)."
Please check if the "12" is a reference number that needs a bracket. In addition, further introduce the question used to measure insomnia (e.g., rating scale).
8. Line 95, kindly explain what kind of brief personal feedback was provided.
9. Line 102-103, kindly indicate whether the maximum likelihood estimation or robust maximum likelihood estimation was used in the analysis.
10. Lines 108-109, "Standardized loading factors (standardized regression weighted estimates) over 0.40 suggest that the indicator variables (items) are significant and representative of their latent variable (PWBI)."
As "significant" is usually used to refer to statistical significance in scientific reports, the sentence may confuse the readers that standardized factor loading greater than 0.40 is statistically significant. Moreover, a citation is needed to support the value of 0.40.
11. Lines 112-114, it is important to present the full picture of the CFA procedure and results. In particular, the authors must present the results for the original unidimensional model first and the steps for adding residual covariance(s), and the results for the modified model for each of the three groups. This information shall be presented before the result of reliability.
12. Lines 118-120, more background information is needed to help readers understand this practice. I wonder if the authors considered examining the receiver operating characteristic curve.
D. Results
1. Line 137, it is hard to follow "three health conditions". The authors may report the results for the generalized anxiety first and then the remaining health conditions to ease understanding.
2. Lines 138-139, CFA results shall be reported first. Moreover, to conduct a group comparison, it is necessary to examine measurement invariance and demonstrate that the PWBI-S is invariant across the three groups and then conduct the latent mean comparison.
3. Table 2, present the results for the model without modification and then the results for the modified model. Also, the model modification (i.e., adding residual covariance) shall be clearly reported in the main text.
4. Figure 1, an asterisk is usually used to report the significance level. It is suggested to just report in the footnote that the factor loadings shown in the figure are standardized values.
5. Lines 190-193, the sentence does not clearly explain why the cutoff of 4-point is not applicable. Kindly revise the sentence to enhance its clarity.
6. Table 3, avoiding using an abbreviation in the title.
E. Discussion
1. The authors are suggested to discuss the underlying reason and implication for adding residual covariances and whether a similar result had also been observed in the past studies.
F. Conclusion
1. Lines 244-245, "Present study demonstrated that PWBI-S is a brief, valid, reliable screening instrument to assess psychological distress in medical students and Mexican healthcare workers"
This is misleading because medical students were not included in the present study.
Reviewer 2 Report
1. The description of the validation of the Spanish version is superficial. Have you made forward- and backward-translations? Is this the first validation, or have there been any validation attempts before?
2. Has sample size been calculated?
3. Expand the acronym MEWS (II 78), PHC (ll 88).
4. What is the rationale for choosing the median PWBI (ll 81-82)?
5. Do the cut-off points actually apply to PHQ2?
6. Square bracket for reference no. 12 (Il 92).
7. What is the rationale for choosing specific questions described in II 83-93. Are these the gold standards according to the authors?
8. Why did the authors choose Cronbach's alpha? It is unclear where the results are shown (in figure 1?)
9. Where does the choice of independent variables come from? Why was there no multivariate regression?
10. Please enter the response rate as a percentage (ll 125).
11. What steps were taken to increase the response rate?
12. Table 1. The number of participants in each group is not correct. Also % of women in the group of physicians is incorrect.
13. Table 1. Please provide cut-off points for the mental state in the footnotes.
14. In table 2, add footnotes and expand abbreviations.
15. Is the total Cronbach's alpha the mean, or was it computed from all components?
16. Figure 1. Individual figures differ (different comparisons), why? The quality of the figure is poor.
17. Please provide results for other cut-off points because the reader does not know where the decision to select a specific result came from.
18. Please expand abbreviations in the footnotes of Table 3.
Round 2
Reviewer 2 Report
The authors addressed all comments.